# SDG102, a H3K36-Methyltransferase-Encoding Gene, Plays Pleiotropic Roles in Growth and Development of Maize (*Zea mays* L.)

**DOI:** 10.3390/ijms23137458

**Published:** 2022-07-05

**Authors:** Yongjian Li, Weifeng Sun, Zhenhui Wang, Chang Wan, Jun Zhang, Xin Qi, Jian Zhang

**Affiliations:** 1Faculty of Agronomy, Jilin Agricultural University, Changchun 130118, China; liyongjian55@163.com (Y.L.); sunweifeng1145@163.com (W.S.); wwan0120@163.com (C.W.); wzhjlau@163.com (Z.W.); zhangjun@jlau.edu.cn (J.Z.); 2Department of Biology, University of British Columbia, Okanagan, Kelowna, BC V1V 1V7, Canada

**Keywords:** maize, histone methylation, flowering time, H3K36, SDG102

## Abstract

Although histone lysine methylation has been studied in thale cress (*Arabidopsis thaliana* (L.) Heynh.) and rice (*Oryza sativa* L.) in recent years, its function in maize (*Zea mays* L.) remains poorly characterized. To better understand the function of histone lysine methylation in maize, SDG102, a H3 lysine 36 (H3K36) methylase, was chosen for functional characterization using overexpressed and knockout transgenic plants. SDG102-deficiency in maize caused multiple phenotypes including yellow leaves in seedlings, late-flowering, and increased adult plant height, while the overexpression of *SDG102* led to reduced adult plant height. The key flowering genes, *ZCN8/ZCN7* and *MADS4/MADA67*, were downregulated in SDG102-deficient plants. Chromatin immunoprecipitation (ChIP) experiments showed that H3 lysine 36 trimethylation (H3K36me3) levels were reduced at these loci. Perturbation of *SDG102* expression caused the misexpression of multiple genes. Interestingly, the overexpression or knockout of *SDG102* also led to genome-wide decreases and increases in the H3K36me3 levels, respectively. Together, our results suggest that SDG102 is a methyltransferase that catalyzes the trimethylation of H3K36 of many genes across the maize genome, which are involved in multiple biological processes including those controlling flowering time.

## 1. Introduction

Flowering time in maize (*Zea mays*) is a crucial agronomic trait that impacts on crop yield and quality. During floral transition, a switch from vegetative to reproductive growth occurs in the shoot apical meristem (SAM). During floral transition, the SAM transforms into an inflorescence meristem (IM) and terminates leaf initiation, with the number of leaves developed being regularly used as a quantitative measurement of the length of the preceding vegetative growth period [1]. In maize, the floral transition is accompanied by the initiation of tassel development in the SAM [2]. The floral transition is controlled by various environmental and endogenous factors that promote flowering including florigen accumulation in maize leaves. Florigen moves from the leaves via the phloem toward the SAM to regulate floral transition [3,4].

FLOWERING LOCUS T (FT), a 23 kDa phosphatidylethanolamine-binding protein, was identified as florigen in *Arabidopsis thaliana* [3,5]. In the SAM, FT interacts with the bZIP transcription factor FLOWERING LOCUS D (FD) to form a complex, triggering the transition to reproductive development by activating floral identity genes such as *APETALA1* (*AP1*) [6]. In the SAM, the main direct target of the FT-FD complex is the MADS-box TF SUPPRESSOR OF OVEREXPRESSION OF CONSTANS1 (SOC1), which is considered as the key integrator of all flowering-inducing pathways in *A. thaliana* [7]. The control of flowering time by the FT-FD module is also conserved in maize (*Zea mays*). The extended family of maize *FT*-like genes are named *Zea mays* CENTRORADIALIS (ZCN), reflecting their diverse functions [8]. To date, three *FT*-like genes have been reported to function as florigen genes in maize. *ZCN8*, a major florigen gene, has been identified as a functional equivalent of Arabidopsis *FT* and is expressed mainly in mature maize leaves [9,10]. A nearly identical paralog, *ZCN7*, has also demonstrated florigenic activity [11]. *ZCN12*, a novel florigen gene, directly controls flowering time together with *ZCN8*. Arabidopsis plants overexpressing *ZCN12* showed an early flowering phenotype [12]. ZCN8 from leaves interacts with DELAYED FLOWERING1 (DLF1), a maize ortholog of Arabidopsis FD, in the shoot apex to form a complex [10,13]. This complex activates the expression of *AP1*-like *MADS-box* meristem identity genes such as *ZmMADS4* (or *ZMM4*), *ZmMADS15* (or *ZMM15*), and *ZmMADS67* (or *ZMM28*), marking the start of maize reproductive development [14,15]. *ZmMADS1*, a maize functional ortholog of the key flowering integrator *SOC1* in *A. thaliana*, is a positive flowering regulator that directly promotes *ZCN8* expression [16,17]. INDETERMINATE1 (ID1), a monocot-specific zinc-finger transcription factor, acts via an autonomous pathway to promote floral transition. It functions upstream of *ZCN8* and *ZCN7* to control their expression, presumably by epigenetic modification of their chromatin structure [9,11,17,18,19]. ZmMADS69 functions as a positive flowering regulator by suppressing the expression of ZmRap2.7, an AP2 transcription factor that functions as a repressor of floral transition via the suppression of *ZCN8/ZCN7* in the aging pathway [20].

Histone protein methylation regulates gene transcription by changing the methylation status of lysine at different sites of histone N-terminal tails and the number of lysine methylations at the same site. Lately, reports on histone methylation in plants mainly focus on lysine residues at positions 4, 9, 27, and 36 of histone tail H3 [21]. Methylation of histone H3 lysine 36 (H3K36) and H3 lysine 4 (H3K4) are most likely associated with the transcription of active euchromatin, while the methylation of histone H3 lysine 9 (H3K9) and H3 lysine 27 (H3K27) are associated with silent chromatin [21]. H3K36 methylation is involved in a variety of processes such as DNA repair, flowering, pollen development, response to stress, and disease resistance in plants [22].

Histone methylation is mediated by histone lysine methyltransferases (HKMTases). Most HKMTases have a conserved Su(var) 3-9, Enhancer of zeste, Trithorax (SET) domain with 130–150 amino acids, and HKMTases are called the Set Domain Group (SDG) proteins. At least five Arabidopsis SDGs (SDG26, SDG24, SDG8, SDG7, and SDG4), six maize (*Zea mays*) proteins (Zmset33, Zmset26, Zmset21, Zmset12, Zmset9, and Zmset4), and five rice (*Oryza sativa*) proteins (SDG736, SDG725, SDG724, SDG708, and SDG707) belong to Ash1 SET proteins that may be involved in H3K36 methylation [23,24]. A total of eight of these have been found to demonstrate HKMTase activity, which plays a crucial role in flowering time and floral organ development. 

SDG8, also known as EARLY FLOWERING IN SHORT DAYS (EFS), CAROTENOID CHLOROPLAST REGULATORY 1 (CCR1), and ASH1 HOMOLOG 2 (ASHH2), is a major di- and tri-methyltransferase, but not a monomethyl transferase for H3K36, and represses flowering by promoting *FLOWERING LOCUS C (FLC)* expression and downregulating the expression of *FT* and *SOC1* [25,26,27]. With regard to controlling the flowering time, SDG8 has also been implicated in the regulation of light and carbon responses, the brassinosteroid pathway, plant defense against fungal pathogens, reproductive organ development, carotenoid composition, and shoot branching [28,29,30,31,32,33]. SDG26, also known as ASHH1, promotes flowering by directly targeting the key flowering gene *SUPPRESSOR OF OVEREXPRESSION OF CONSTANS1/AGAMOUS-LIKE 20 (SOC1/AGL20)*, and is essential for producing histone H3 lysine 4 trimethylation (H3K4me3) and histone H3K36me3 at the chromatin of this gene [34]. A *SDG8 SDG26* double mutant showed a phenotype similar to SDG8 [early flowering by reduced expression of *FLC* and *MADS AFFECTING FLOWERING (MAF),* and increased the expression of *FT* and *SOC1* as well as reduced plant body size], but opposite to SDG26 (late flowering by the increased expression of *FLC* and *MAF5*, and reduced expression of *FT* and *SOC1*, with an increased rosette size), suggesting that the SDG8 mutation is epistatic to the SDG26 mutation [35]. SDG4 is involved in the regulation of stamen development and pollen tube growth via H3K4 and H3K36 methylation. *SDG4* knockout resulted in the suppression of pollen tube growth. However, *SDG4* overexpression led to growth restraints, specifically anther degeneration and male sterility [36,37]. In addition, *SDG4* knockout mutants exhibited reduced lengths of the root apical meristem and primary root by H3K36 methylation [38]. SDG7 is required for the proper timing of the vernalization process. Loss of SDG7 function represses *FLC* expression by increasing both H3K27me3 and the levels of long non-coding RNAs, even without cold exposure [39]. 

SDG725, which encodes a rice homolog of SDG8, is also a H3K36me2 and H3K36me3 methyltransferase. Interestingly, SDG725 promotes brassinosteroid-related gene expression and rice flowering by depositing H3K36me2/me3 at the chromatin of flowering genes, in contrast to the function of SDG8 [40,41]. SDG708, the closest rice homolog of SDG26, is a mono-, di-, and tri-methyltransferase for H3K36. SDG708 contributes to flowering under both short-day and long-day conditions by activating the key flowering genes *Heading date 3a* (*Hd3a*), *FLOWERING LOCUS T1 (RFT1*)*,* and *Early heading date 1* (*Ehd1*) in rice (*Oryza sativa*) [42]. In addition to controlling the flowering time, SDG708 also acts as a positive regulator of drought tolerance. The overexpression of SDG708 increased the grain yield and drought tolerance under normal and drought stress conditions in rice [43]. The overexpression of rice *SDG736* in *A. thaliana*, the closest rice homolog of Arabidopsis SDG4, resulted in early flowering. The key flowering gene *FLC* was repressed while *SOC1* was overexpressed [44]. SDG724, which has the highest sequence similarity to Arabidopsis SDG7 and maize SDG110 (or Zmset33), also contributes to rice flowering by regulating the flowering genes *MADS*-*box gene 50* (*OsMAD50*) and *RFT1*. Loss-of-function of *SDG724* showed reduced H3K36me3, H3K36me1, and particularly H3K36me2 levels in vivo [45].

SDG102 is the closest maize homolog of both rice SDG708 and Arabidopsis SDG26 [46]. Plants deficient in SDG102 exhibited more DNA damage than WT plants after UVB irradiation [47]. However, there have been no reports regarding the functions and roles of SDG proteins in maize growth and development due to the lack of mutants with obvious phenotypes.

In this study, we demonstrated that SDG102 is an H3K36 methyltransferase and show that SDG102 promotes flowering by activating the key flowering genes *ZCN8/ZCN7* through decreased H3K36me2 and H3K36me3 levels in the chromatin regions of these genes. In addition, we show that SDG102 participates in the regulation of numerous other processes including leaf color, plant growth, and plant height. 

## 2. Results

### 2.1. SDG102 Phylogenetics and Sequence Alignment

Phylogenetic analysis of the Ash1 SET domain proteins indicated that maize SDG102 is closely associated with rice SDG708 and Arabidopsis SDG26/ASHH1 (Appendix A). These three proteins contain three identical domains, namely AWS (associated with SET), SET [Su(var) 3-9, E(z), Trithorax], and post-SET domains, and the molecular weights of the three proteins were similar (Appendix A). In addition, the three proteins exhibited high similarity according to the sequence alignment analysis. The identical amino acid sequences between SDG102 and SDG26 or SDG708 were 76.2% or 87.5%, respectively (Appendix A). Both SDG708 and SDG26/ASHH1 have previously been reported to function as H3K36-methyltransferase in rice (*Oryza sativa*) and *Arabidopsis thaliana*, respectively [34,42]. Thus, that SDG102 might methylate H3K36 in maize became a testable hypothesis in this study. 

### 2.2. Generation of SDG102 Downregulation and Overexpression Lines

Two experimental approaches were used to clarify the function of SDG102 in maize. The first approach was using RNA interference to generate transgenic plants with downregulated *SDG102*. One of six independent transgenic plants with the highest SDG102 downregulation (AS1) was selected for further analysis, as shown in Figure 1a. The second approach employed the CRISPR/Cas9 method, which was carried out to generate *SDG102* knockout transformants. One transgenic maize plant (AS3), with one base deletion located in the seventh exon, was selected for further studies (Figure 1c). Southern blot analysis confirmed that the AS3 line also contained one independent insertion of the transgene within the whole genome (Figure 1d).

Subsequently, transgenic maize plants overexpressing (OE) *SDG102* under constitutive 35S promoter were generated by pollen-mediated transformation. Two independent OE lines (OE2 and OE3) were selected for further studies (Figure 1b). Southern blot analysis indicated that these two OE lines contained only one independent insertion of the transgene within the whole genome (Figure 1d).

### 2.3. Mutation Types and Editing Rate of CRISPR/Cas9

A protoplast assay was performed to verify the editing efficiency of the two constructed CRISPR/Cas9 vectors that targeted the third and seventh exons of the SDG102 gene, respectively, prior to transformation. Direct sequencing of PCR products encompassing the targeted sites indicated that there were double peaks at 3–5 bp upstream of PAM, which meant that more than one PCR-amplified DNA template sequences existed and the genomic DNA sequence was edited (Figure 2a,b). Purified PCR products were cloned into pMD19-T, and 20 single clones from each target site were sent for Sanger sequencing. The sequencing results indicated that the deletion of one to six nucleotides was the main mutation type, especially the deletion of one or two nucleotides (Figure 2c–e). No insertion mutation type was detected, likely due to the limited sample size. 

The plasmid pCAMBIA3301 containing the Green fluorescent protein (GFP) gene was used to transform the maize protoplasts simultaneously to calculate the transformation rate. The transformation rate refers to the ratio of the number of light-emitting cells to the total number of cells. In this study, the transformation rate was 60% (Appendix A). The editing rates of the two small guide RNAs (sgRNA) were 33% and 56%, respectively [editing rate = numbers of mutation events/(numbers of individual amplicon-containing plasmid clones ∗ transformation rate); Appendix A]. The factors affecting the Cas9 targeting efficiency included the binding of a sgRNA to its target site (GC-content) and secondary structures of target sgRNAs [11]. The results suggested targets with high GC-content had higher editing efficiency than those with low GC-content (Appendix A).

### 2.4. Phenotypic Alteration in the Down- and Upregulation Lines of SDG102 Expression

Phenotypic analysis was conducted on BC4-F3 or F3 progenies of homozygous transgenic plants and WT segregant and WT plants, revealing that the perturbation of *SDG102* expression causes several developmental defects. Seedlings of both AS1 and OE3 plants germinated normally, but the growth of transgenic plants was slower during the vegetative phase than the WT plants (Figure 3a). AS1 plants produced yellow leaves during the seedling stage. Interestingly, AS3 plants generated more yellow leaves with larger surface areas than AS1 plants (Figure 3b,c). The leaf color of the AS1 and AS3 plants remained yellow at the vegetative-to-reproductive phase transition stage (i.e., V6/V7, which are referred to the seedlings with the sixth or seventh leaf presence, fully extended collars, and the eighth/ninth leaves visible; Figure 3d,e). However, the leaf color of AS1 plants returned to normal and the bottom leaves of the AS3 plants remained yellow at the post-transition (V8) stage (Figure 3f,g). The leaf color of the AS3 plants returned to normal up to the V10 stage, despite a few yellow spots remaining on the bottom leaves (Figure 3h). At the V6 stage, pigment content in yellow leaves of the AS lines showed decreasing to approximately 30–32% of the WT plants (Figure 3i). Once yellow leaves disappeared (i.e., when the plants reached the adult stage and turned completely green), the pigment levels of the AS lines were like those of the WT (Figure 3j). Previous reports have demonstrated that the internal structures of chloroplasts in the yellow sections of leaves are severely disaggregated, which results in a slow growth rate [48]. 

Floral transition can be estimated based on the extent of apex elongation. The vegetative meristem is more symmetrical in shape, whereas the reproductive meristem is much taller than it is wide. No obvious differences were observed for the SAM of the *SDG102* downregulated plants compared with the WT before the floral transition (Figure 3k). The meristem of the WT plant was much taller compared to the width, but the meristem of the AS line remained symmetrical in shape at the V5 stage, indicating that the floral transition of the WT occurred much earlier than that of the AS line (Figure 3).

At the floral transition stage of WT, the fully extended leaf number of the AS lines was one less and the plant height was approximately 16 cm shorter than that of the WT plant (Figure 3f; Table 1). However, although the plant height of the OE lines was 14 cm shorter than that of the WT plant, the number of fully extended leaves of the OE lines was like that of the WT plant (Table 1). The downregulation of *SDG102* expression in the AS lines resulted in delayed flowering (Figure 3m), and adult plants of the AS lines had much taller plant height, ear height, and a larger leaf number than the WT (Table 1). In addition, the ratio of ear height to plant height showed a statistically significant difference between the AS lines and the WT, indicating that the plant height increase in the AS lines is mainly due to an increase below the ear placement (Table 1). In contrast, adult plants of the OE line had shorter plant and ear heights, but no statistically significant differences in the flowering time and leaves number were observed compared to those of the WT plants (Table 1). Interestingly, the cob color of the OE line changed from red to white (Figure 3n).

Together, the phenotypic alterations described above were observed in more than one AS and OE line, suggesting that SDG102 plays a key role in the development of maize. Therefore, the AS1 and OE3 lines were chosen as representatives of the two classes of the SDG102 transgenic plants and used for further molecular analysis.

### 2.5. SDG102 Is Required for Normal Expression of a Set of Genes Involved in Various Biological Processes

RNA-Seq analyses were performed on plants for both the down- and upregulation of SDG102 expression and WT to better address the pleiotropic effects of the perturbation of SDG102 expression and identify genes and biological processes controlled by SDG102. Through alignment with version V4 of the B73 genome, 28,093 and 27,852 expressed annotated transcripts were detected in the AS1 and OE3 plants, respectively. Among them, 1079 AS1 and 1167 OE3 genes showed differential expression in comparison with the WT (|log_2_ foldchange|>2 and false discovery rate (FDR) <0.05; Appendix A). More than half of these differentially expressed genes (DEGs), 680 in AS1 and 634 in OE3, showed decreased expression levels during floral transition. These results revealed that SDG102 is associated with both the gene activation and repression in maize growth and development. The GO term enrichment analysis of these DEGs revealed that SDG102 modulates genes involved in multiple biological processes (Appendix A). In AS1 plants, this revealed the downregulation genes involved in the hormone-mediated signaling pathway, the response to acidic chemicals, the cellular response to endogenous stimuli, and the cellular response to hormonal stimuli. In OE3 plants, significant GO term enriched genes included serine-type exopeptidase activity, acid phosphatase activity, and serine-type carboxypeptidase activity. AS1 plants had 152 significant GO terms downregulated and 44 significant GO terms upregulated, respectively, when the DEGs were identified using FDR < 0.05. In contrast, the OE3 plants had 132 significant GO terms upregulated and 48 significant GO terms downregulated. The same trend was observed when identifying DEGs with the criteria of |log_2_ fold change |≥2 and FDR < 0.05 (Appendix A). These data revealed that SDG102 downregulation in maize had a greater impact on downregulated DEGs than upregulated DEGs, and SDG102 overexpression had a greater impact on the upregulated DEGs. This implies that SDG102 mainly plays a role in promoting gene expression.

### 2.6. SDG102 Promotes Flowering by Regulating H3K36 Methylation of Flowering Genes

The SDG102-deficient plants displayed a late-flowering phenotype, which prompted us to examine the expression of known maize flowering regulatory genes. RT-qPCR indicated that the key flowering-related genes ZCN8/ZCN7 and MADS4/MADS67 were downregulated in AS1 plants (Figure 4a). The FT-like ZCN8 and its close paralog ZCN7 are key genes in controlling the flowering time in maize. Their proteins are produced in leaves and move to the shoot apex to promote MADS4 and MADS67 by interacting with DLF1, ultimately regulating the flowering time [9,10,11,15]. To obtain insights into the molecular mechanism underlying the downregulation of flowering genes in SDG102-deficient plants, ChIP assays were conducted to investigate the histone methylation levels in the ZCN8/CZN7 chromatin. The AS1 line showed reduced H3K36me2 and H3K36me3 levels in most regions of the ZCN8/ZCN7 chromatin unlike the WT (Figure 4b,c), which indicates that the downregulation of the key flowering genes is caused by the reduced methylation status of H3K36. 

Collectively, SDG102 regulates the expression of flowering genes by controlling the H3K36 methylation levels at these gene loci, finally controlling the flowering time in maize.

### 2.7. SDG102 Is a Broad Trimethyl Transferase on H3K36

To evaluate the impact of SDG102 on global histone methylation patterns, Western blotting and ChIP-sequencing (ChIP-seq) using H3K36me3 antibodies were performed. The experiments were conducted on the ninth leaves collected from the V7 stage plants to compare the SDG102-knockdown and overexpression plants with the WT. Western blotting analysis indicated that the H3K36me3 levels were markedly reduced in AS1 and increased in OE3 compared to the levels in the WT (Figure 5a). The same trends were observed in the ChIP-seq analysis (Figure 5b). Interestingly, H3K36me3 in the WT, OE3, and AS1 was most abundant within the gene body regions near the transcription start site (TSS) regions. There were 22,351 gene loci in OE3, 18,799 gene loci in the WT, and 9059 gene loci in AS1 with H3K36me3, respectively (Figure 5c). 

We found 6922 genes in the AS1 plants and 6708 genes in the OE3 plants with significantly altered H3K36me3 compared to those in the WT (|log2 foldchange| >2). Among them, 82% of genes in AS1 and 73% in OE3 showed downregulated and upregulated H3K36me3, respectively. GO term enrichment was performed on these genes with significantly altered H3K36me3. In OE3, this revealed the upregulated H3K36me3 of genes involved in chromatin organization, histone modification, covalent chromatin modification, and nucleoplasm (Appendix A). In AS1, significant GO terms included nucleoplasm, protein localization to organelles, and purine nucleoside binding (Appendix A). Interestingly, a total of 176 significant GO terms were identified for genes with downregulated H3K36me3 in AS1, but no significant GO terms were identified for genes with upregulated H3K36me3. However, a total of 130 significant GO terms were found for genes with upregulated H3K36me3 in OE3, but only 19 GO terms were found on genes with downregulated H3K36me3 (Appendix A).

Collectively, SDG102 overexpression in maize dramatically increased H3K36 me3 on many gene loci, whereas SDG102-deficient plants caused reduced H3K36 me3 on multiple gene loci. Therefore, SDG102 serves as a broad H3K36me3 in maize.

## 3. Discussion

In this study, we characterized the role of SDG102, a novel H3K36 methyltransferase gene, showing that it is essential for proper maize growth and development. Loss-of-function and overexpression analyses suggest that SDG102 participates in the regulation of several key biological processes including leaf color, plant growth, flowering time, and plant height. Remarkably, SDG102 deficiency in maize led to a dramatic reduction in H3K36me3 levels, as evidenced at the genome level by both Western blotting and ChIP-seq analysis as well as at key flowering regulatory gene loci, as shown by the ChIP-PCR analysis. More importantly, the overexpression of *SDG102* increased the H3K36me3 levels. Therefore, we can conclude that SDG102 is a broad H3K36 trimethyl transferase. 

Previous studies have demonstrated that eight proteins from *A. thaliana* and rice (*Oryza sativa*) are capable of catalyzing the methylation of H3K36 [23,24]. SDG102 is a global H3K36 methyltransferase that is distinct from its Arabidopsis homolog SDG26. Loss-of-function *sgd26* mutants had reduced H3K36 methylation levels at specific gene loci but did not exhibit a genome-wide reduced H3K36 methylation level [25,34]. However, SDG708, the closest rice homolog of SDG102, is also a broad H3K36 methyltransferase, which seems to imply that SDG proteins are more conserved between maize and rice [42]. In addition, Arabidopsis SDG8 and rice SDG724 are broad H3K36 methyltransferases, but Arabidopsis SDG7 and rice SDG725 only alter H3K36 methylation levels at specific genes [15,25,39]. 

*ZCN8/ZCN7* are key flowering regulatory genes in maize, which are homologous to florigen genes *Hd3a* and *RFT1* in rice (*Oryza sativa*) and *FT* in *A. thaliana*. Their proteins move from the leaves to the shoot apical meristem via the phloem, where they function in floral transition [9,11]. In the SDG102-deficient plants, the expression of *ZCN8* and *ZCN7* was downregulated and reduced H3K36me3 levels were observed at these two chromatic loci, which implies that SDG102 inhibited the expression of *ZCN8* and *ZCN7* by reducing the H3K36me3 levels and led to a late-flowering phenotype. This result is consistent with previous studies that indicated that control of the flowering time of both temperate and tropical maize is associated with chromatin modifications [11]. 

Of the 10 Ash1 family protein members of *Arabidopsis* and rice, six have been identified to be involved in the flowering time control. These are SDG7, SDG8, and SDG26 in *A. thaliana*, and their corresponding homologs in rice, *SDG724*, *SDG725*, and *SDG708*. Arabidopsis *sgd26* mutants displayed a late-flowering phenotype and decreased H3K36me3 and H3K4me3 levels at the chromatin of *SOC1* [34]. *ZmMADS1* is a homolog of *SOC1* in maize [16], but its expression is unaffected in SDG102-deficient plants. Rice *SDG708*-knockdown mutants displayed a late-flowering phenotype due to the downregulation of H3K36me2/3 levels at the chromatin of the key flowering regulatory genes *Hd3a*, *RFT1*, and *Ehd1* [42]. This indicates that histone methyltransferases in both the dicots and monocots have their own characteristics and are evolutionarily conserved. Arabidopsis SDG7 is a negative regulator of vernalization, and the loss of *SDG7* produces a partially vernalized state without cold exposure. Moreover, the loss of *SDG7* results in increased H3K27me3 deposition on the *FLC* chromatin, which shows an early flowering phenotype by suppressing *FLC* expression [39]. The loss-of-function of *SDG8* in *Arabidopsis* results in reduced H3K36me2/3 and increased H3K36me1 at the *FLC* chromatin, which causes reduced *FLC* expression and early flowering [27,35]. The loss-of-function of *SDG725* and *SDG724* results in late flowering in rice. SDG725 is mainly required for H3K36me2/3 deposition at flowering genes such as *Ehd3*, *Ehd2*, *OsMADS50*, *Hd3a*, and *RFT1* [41], whereas SDG724 is necessary for H3K36me2 deposition at *OsMADS50* and *RFT1* [15,45]. *OsMADS50* is the closest homolog of Arabidopsis *SOC1* in rice [49]. These results indicate that the ASH1 protein family controls flowering in *Arabidopsis* and rice by regulating H3K36 histone methylation at key flowering genes. However, SDG102 promotes flowering in maize by depositing H3K36me2/3 at the chromatin of the key flowering gene *ZCN8/ZCN7*. 

In fact, the function of SDG102 is not limited to the flowering time regulation. *SDG102* downregulation caused multiple phenotypes in addition to late flowering including yellow leaves and a slow seedling growth rate, and increased plant height. In contrast, overexpression led to reduced plant height. These results indicate that SDG102 participates in multiple biological processes during early plant growth. According to RNA-Seq analysis, SDG102 is necessary for the proper expression of numerous genes involved in diverse biological processes. In addition to flowering, ASH1 family proteins in rice and *Arabidopsis* such as SDG26 and SDG8 in *Arabidopsis* and SDG708 and SDG725 in rice also affect a variety of plant phenotypes reported in previous studies [28,29,30,33,40,43]. RNA-Seq analyses demonstrated that DEGs between the *sdg102* mutants and the WT are distributed across functional categories such as hormone signal transduction, cell composition of the nucleosome and histone modification, carbohydrate metabolism, and response to stress. These data provide clues for further exploration of the function and molecular mechanism of H3K36 methylation mediated by SDG102. 

We analyzed the global genic distribution profiles of H3K36me3 in maize using ChIP-seq analysis. The H3K36me3 pattern revealed in this study is consistent with that found in *A. thaliana* and rice [50]. Moreover, mutant studies were performed to uncover genome-wide H3K36 methylation patterns in *SDG102*-overexpressing or deficient plants, thereby increasing our understanding of the role of SDG102 in H3K36me3 formation. H3K36me3 was enriched in gene body regions and was especially abundant near the TSS regions, similar to what has been described for *A. thaliana* and rice [50]. The genome-wide decrease in the H3K36me3 levels is very remarkable in SDG102-deficient plants, both near the TSS and in other regions of the gene body. Moreover, a substantial genome-wide increase in the H3K36me3 levels was also observed in the *SDG102*-overexpressing plants. The level of H3K36me3 in the overexpressing plants was significantly higher than that of the downregulated plants. Interestingly, previous studies have shown that a genome-wide decrease in H3K36me3 levels is visible but not significant in the mutants of both Arabidopsis *SDG8* and rice *SDG708*. In particular, H3K36me3 level decreases were found only near the TSS in rice *SDG708*-deficient plants. However, decreases in the H3K36me3 level were observed in gene body regions, except for TSS in the Arabidopsis *SDG8*-deficient mutant [42]. According to previous reports, global changes in H3K36 methylation were not clearly detected by Western blotting in mutants of both Arabidopsis *SDG8* and rice *SDG708* [25,42]. However, in this study, global level changes in the H3K36 methylation were clearly detected by Western blotting in *SDG102*-deficient or overexpressing plants, highlighting its central role in the global regulation of gene expression in maize.

## 4. Materials and Methods

### 4.1. Plant Materials and Genetic Transformation

DNA fragments including nucleotides 831–1522 of the *SDG102* transcript were amplified as inverted repeats using the primers listed in Appendix A. The inverted repeats were cloned into vector pB7GWIWG2, flanked by a spacer sequence, yielding *P35S::SDG102-RNAi* (Appendix A) [51]. The 12-day-old HiII immature embryos were bombarded with 5 mg of gold particles coated with 2 μg of plasmid DNA for co-transformation [52]. Six independent transgenic lines overexpressing the *SDG102* antisense transcript (AS lines) were obtained. One transgenic line, with the highest reduction in *SDG102* RNA levels, was backcrossed four times to WT B73 to minimize the genetic background influences, and then self-fertilized at least twice to obtain homozygous lines used for molecular and phenotypic analysis (BC4-F3 generation; Appendix A).

The full-length *SDG102* cDNA sequence was cloned into *pCAMBIA3301* with the 35S promoter to obtain an overexpression vector using the primers listed in Appendix A. The sgRNA-Cas9 binary expression vector was generated using the following procedure: Two sgRNA sequences were selected within the *SDG102* gene, targeting the third exon and the seventh exon, respectively (http://cbi.hzau.edu.cn/cgi-bin/CRISPR (accessed on 9 May 2022), target sequence in Appendix A). The secondary structure analysis of the sgRNA sequences was performed using the program RNA Folding (http://unafold.rna.albany.edu/?q=mfold/RNA-Folding-Form2.3, accessed on 9 May 2022) (Appendix A). Subsequently, these two sgRNAs were cloned into the *pBUE411* of the CRISPR/Cas9 system [33]. These three plasmids were employed to transform B73 directly via pollen-mediated transformation [53]. Three overexpression transgenic lines of *SDG102* and seven knockout lines for *SDG102* were obtained using *CRISPR/Cas9*. T0 plants were self-pollinated twice to obtain homozygous transgenic lines, and two overexpression events with the highest expression (OE2 and OE3) and one CRISPR knockdown event (AS3) were selected for further analysis.

### 4.2. Transient Expression in Maize Protoplast

Protoplasts were obtained from 15-day-old leaves of B73 and transformed to verify the editing efficiency of two sgRNA-Cas9 vectors, according to a previously described method [54]. Plasmid DNAs were supplied using the EndoFree Plasmid Maxi Kit (Qiagen, Hilden, Germany) and 200 µL maize protoplasts (at a concentration of 2 × 105 cells mL^−1^) were transformed with 20 µg plasmid DNA, according to a previously described procedure [55]. After incubation in W5 solution for 12 h, the isolated protoplasts were transferred into W5 solution containing 660 mM mannitol for 1 h [56]. The protoplasts were subsequently collected by centrifugation at 100× *g* for 3 min for mutation detection.

### 4.3. Mutation Detection

Genomic DNA was extracted from the transformed maize protoplasts using the CTAB protocol. PCR amplifications were performed using primer pairs encompassing the target sites (Appendix A). Purified PCR products were cloned into pMD19-T to carry out mutation analysis, and 20 single clones from each target site were sent for Sanger sequencing.

### 4.4. Phenotypic and Morphological Analysis

Phenotypic analyses were performed on plants grown in the field. Homozygous transgenic plants and WT B73 were used to collect data. More than 40 plants for each genotype were monitored to determine the plant traits and flowering time. The vegetative growth stage was defined on the basis of the leaf collar appearance of the top leaf [57]. Data collection with respect to the total leaf number was performed by marking the sixth and twelfth leaves. Plant height was measured from the ground level to the top of the tassel. Flowering time was recorded in the field.

### 4.5. Measurement of Photosynthetic Pigments

Fresh leaf tissue (0.2 g) was sampled in triplicate from the downregulated transgenic and WT plants. Pigments were isolated using 95% ethanol. Chla, Chlb, and carotenoid contents were determined using a UV VIS-1 spectrophotometer by measuring the absorbance at 665, 649, and 470 nm, respectively. Pigment content levels were calculated based on a previously described method [58].

### 4.6. Gene Expression Analysis

Fresh leaves from maize plants at the V6/V7 stage corresponding to the floral transition were harvested for total RNA extraction using an EasyPure Plant RNA Kit according to the manufacturer’s instructions (Transgen Biotech, Beijing, China). A total of 2 ug RNA was prepared for first-strand cDNA synthesis using EasyScript One-Step gDNA Removal and cDNA Synthesis SuperMix, following the manufacturer’s protocol (Transgen Biotech). RT-PCR was performed using the gene-specific primers listed in Appendix A. The housekeeping gene ZmACT1 (GRMZM2G126010) was used as an internal control to normalize the data. The data shown in the bar charts represent the mean values of three replicates.

### 4.7. RNA Sequencing

Transcriptome analyses were conducted on the total RNA isolated from mature ninth leaves collected from V7 stage plants (three biological replicates, with five plants pooled for each replicate per genotype). Paired-end sequencing libraries were prepared using a TruSeq RNA Sample Preparation Kit v2 (Illumina, San Diego, CA, USA) and sequenced on a HiSeq2000 (Illumina), developing 40 to 46 million reads per replicate. The raw reads were processed and filtered to eliminate poor-quality reads, and finally, approximately 6 Gb of high-quality 125-bp paired-end reads were produced from each library. The RNA-Seq clean reads were aligned to the maize genome (B73_RefGen_v4, http://plants.ensembl.org/Zea_mays/ Info/Index, accessed on 9 May 2022). A high degree of correlation between replicates (R > 0.95) suggested that high quality RNA-Seq data were obtained. In addition, data quality analysis was performed by visualizing the results of the principal component analysis (Appendix A). The read count levels were normalized to the reads per kilobase per million reads. DEGs were defined with an absolute value of |log2 fold change| ≥2 and FDR < 0.05. The identification of GO categories significantly enriched within DEGs was performed using AgriGO2 [59].

### 4.8. ChIP Assays

Fresh mature leaves at the V6/V7 stage were used to perform ChIP assays, as previously described [60]. The antibodies used in this study were anti-H3K36me2/3 (ab9049, cst4909). Quantitative PCR was performed to determine the enrichment of the immunoprecipitated DNA. The sequences of the primers used for the ChIP assays are listed in Appendix A.

### 4.9. ChIP-Seq 

ChIP-seq libraries were constructed using the NEBNext Ultra DNA Library Prep Kit for Illumina (New England Biolabs, Ipswich, MA, USA). Subsequently, they were sequenced on the HiSeq2000 (Illumina), and reads were mapped to the maize genome (B73_RefGen_v4, http://plants.ensembl.org/Zea_mays/Info/Index, accessed on 9 May 2022) using Bowtie [61]. Peak calling was carried out separately for each biological replicate unlike in the pre-blood control using MACS (Zhang et al., 2008), with a *q*-value < 0.05.

## Figures and Tables

**Figure 1 ijms-23-07458-f001:**
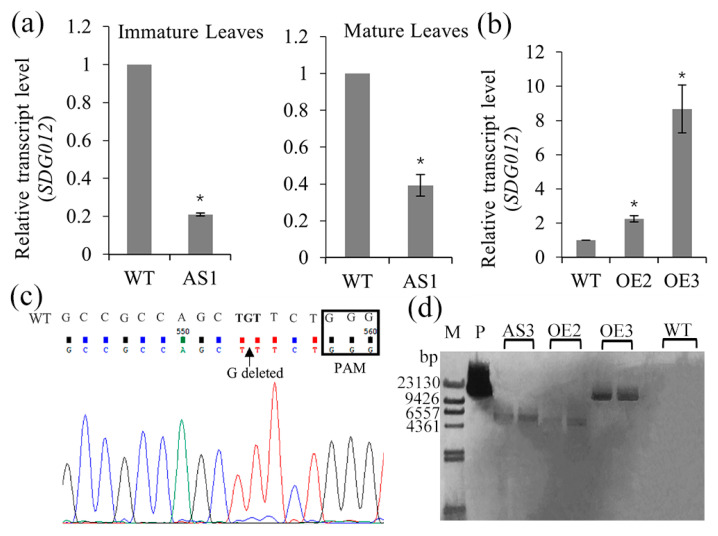
The downregulation and upregulation of *SDG102* expression in transgenic maize plants. (**a**,**b**) Relative transcript level of *SDG102* in the wild-type (WT), *SDG102-RNAi* line (AS1; (**a**)), and overexpression line (OE2 and OE3; (**b**)). Asterisks (*) indicate statistically significant differences between the indicated genotypes and the wild type (*p* < 0.001). Error bars indicate standard deviation (*n* = 3). *ZmACT1* was used as an internal control. (**c**) Sequencing results of CRISPR/Cas9-mediated mutations on the *SDG102* target site in transgenic plants (AS3). WT represents the sequence of the wild-type. Sequence inside the rectangle is the Protospacer Adjacent Motif (PAM) sequence. (**d**) Southern blot analysis using DNA extracted from WT, AS3, OE2, and OE3 (M = molecular weight marker; *p* = probe bar gene).

**Figure 2 ijms-23-07458-f002:**
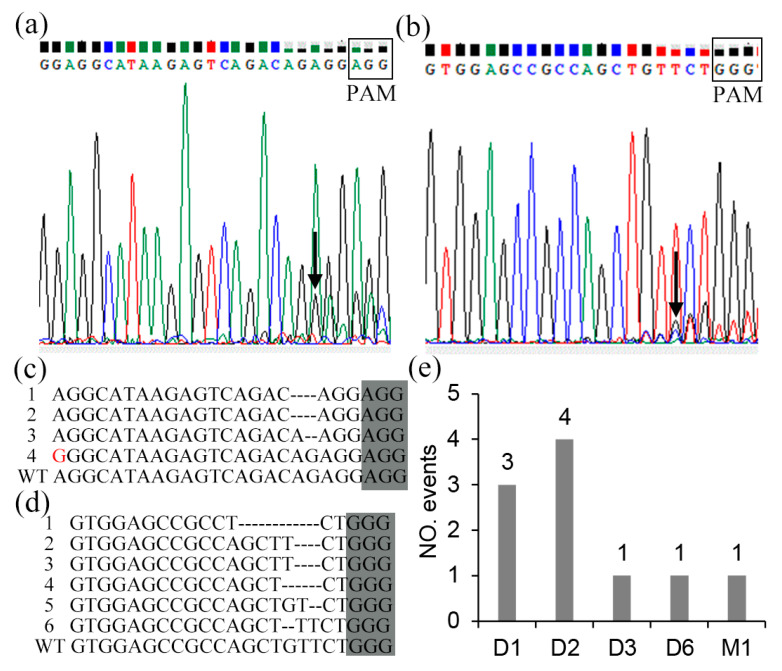
The detection of the CRISPR/Cas9-mediated mutation on the SDG102 target site in protoplast, arrows indicated genomic DNA sequence was edited. (**a**,**b**) Direct sequencing of PCR products containing the targeted sites of the third exon and seventh exon, respectively. (**c**,**d**) Monoclonal mutation analysis of the targeted sites of the third exon and seventh exon, respectively. (**e**) Mutation-type analysis (D1 = 1 bp deletion; D2 = 2 bp deletion; D3 = 3 bp deletion; D6 = 6 bp deletion; M1 = nucleotide replacement mutation).

**Figure 3 ijms-23-07458-f003:**
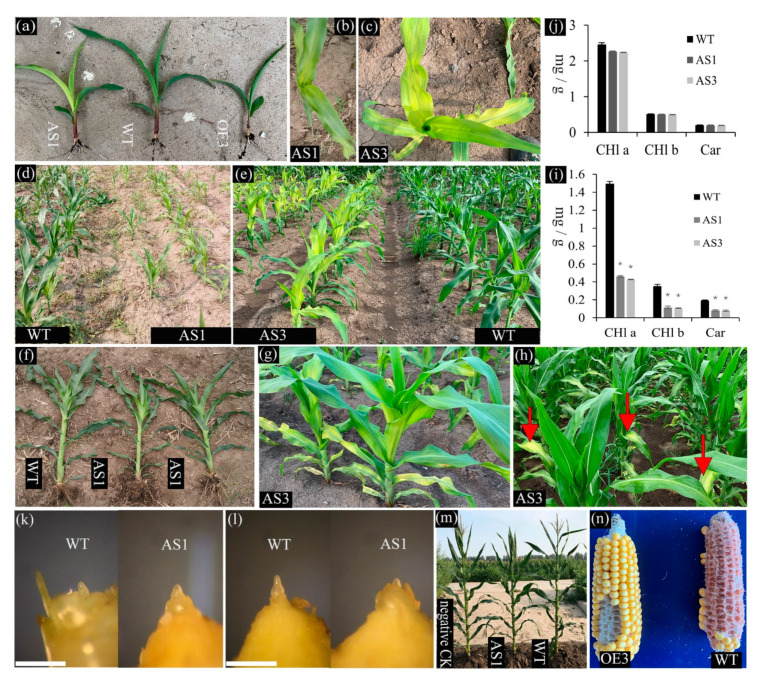
The perturbation of SDG102 expression induces pleiotropic effects on plant development. (**a**–**h**) Examples of visible phenotypes in the down- (AS1 and AS3) and upregulation (OE3) lines of *SDG102* expression compared with the wild-type (WT), including the slow growth rate at seedling stage (**a**), yellow leaves in AS1 (**b**,**d**) and AS3 (**c**,**e**) before V8 (the seedlings with eight leaves of fully extended collars). Leaves color of AS1 returned to normal (**f)** after V8. But the bottom leaves of AS3 remained yellow after V8 (**g**) and they returned to normal until the V10 stage despite a few yellow spots remaining on the bottom leaves (**h**). (**i**,**j**) Photosynthetic pigment content in leaves of the WT and *SDG102* downregulation expression plants (AS1 and AS3). Chlorophylls (Chl a and Chl b) and carotenoids (Car) were measured in acetone extracts from the second top leaf at V6 (**i**) and R1 (silking stage; (**j**)) growth stages. Values based on fresh weight and given as the mean ± standard deviation of three independent determinations. Asterisks (*) indicate statistically significant differences between the indicated genotypes and the WT by the Student’s *t*-test analysis (*p* < 0.01). (**k**,**l**) Morphological changes of maize shoot apices collected from V4 stage plant (**k**) before and V5 stage (**l**) during the floral transition of the WT and *SDG102* downregulation expression plants (AS1). Scale bars (**k**,**l**) = 1 mm. (**m**) AS1 displayed a late-flowering phenotype. (**n**) Cob color of OE3 changed from red to white. Negative CK = wild type segregants.

**Figure 4 ijms-23-07458-f004:**
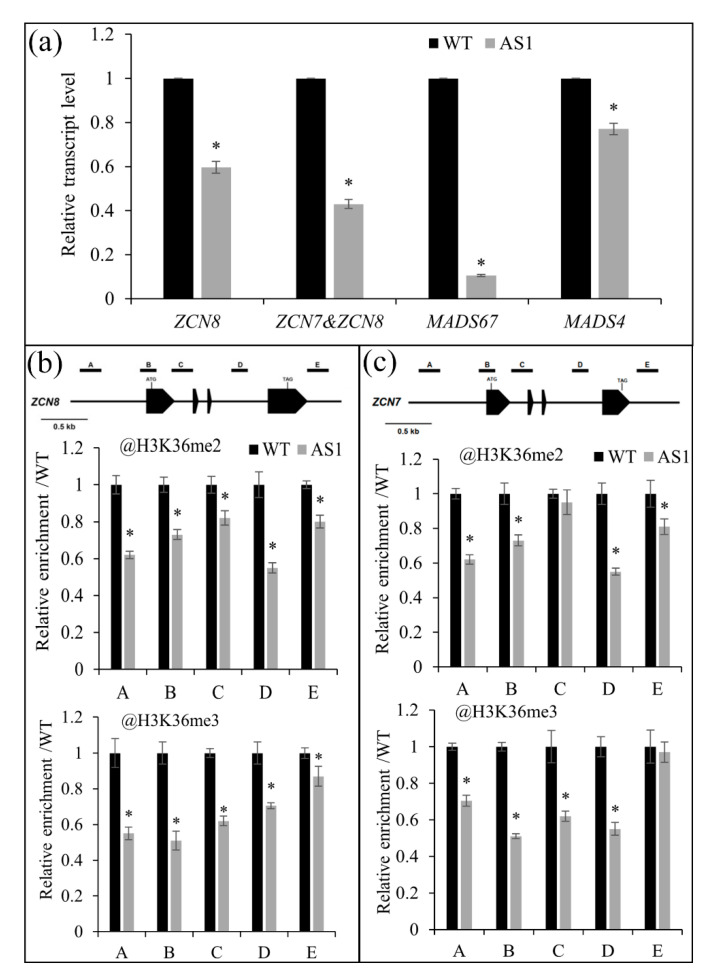
The relative transcript level and ChIP analyses at the chromatin of key flowering genes. (**a**) The relative transcript level of key flowering genes in the wild-type (WT), *SDG102-RNAi* line (AS1). (**b**,**c**) Schematic representation of the *ZCN8* and *ZCN7* genes, with black boxes representing exons. ChIP assays were performed on the indicated regions using antibodies against H3K36me2 and H3K36me3. The values were normalized to *ZmACT1* and are relative to that of WT. Asterisks (*) indicate statistically significant differences between the indicated genotype and the WT (*p* < 0.01).

**Figure 5 ijms-23-07458-f005:**
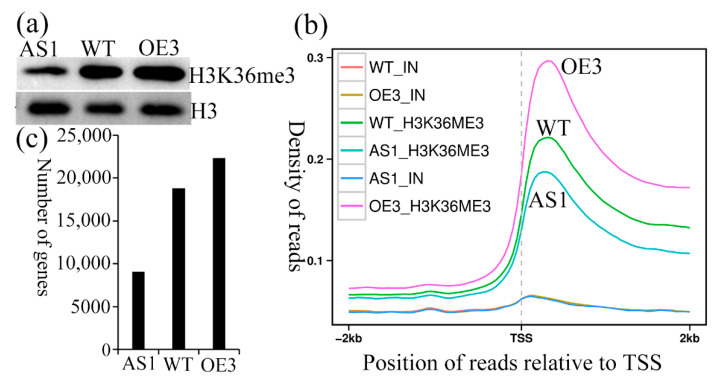
The perturbation of the *SDG102* expression induced global changed H3K36me3 in *SDG102* knockdown (AS1) and overexpression (OE3) plants vs. the wild-type (WT). (**a**) Western blot analysis. (**b**) The gene aggregation plots for H3K36me3. (**c**) The total number of genes with H3K36me3. TSS = transcription termination site.

**Table 1 ijms-23-07458-t001:** The phenotypic variations of the AS and OE lines.

	Jointing Stage		Flowering Time (Days)	Adult Plant			
	PHT (cm)	Number of Leaves		PHT (cm)	EHT (cm)	Ratio of EHT/PHT	Number of Leaves
WT	48.5 ± 4.3	6.9 ± 0.2	63.3 ± 2.0	206.6 ± 9.1	84.1 ± 7.7	0.41 ± 0.04	21.9 ± 0.7
AS1	32.1 ± 4.7 *	5.8 ± 0.4 *	69.1 ± 2.7 *	213.6 ± 8.6 *	95.1 ± 8.2 *	0.45 ± 0.04 *	22.6 ± 0.8 *
AS3	31.1 ± 4.0 *	5.3 ± 0.5 *	70.0 ± 2.6 *	225.3 ± 9.7 *	100.9 ± 9.5 *	0.45 ± 0.03 *	22.8 ± 0.8 *
OE3	34.5 ± 4.5 *	6.5 ± 0.5	62.3 ± 1.5	176.3 ± 15.0 *	71.3 ± 8.5 *	0.41 ± 0.03	21.5 ± 0.7
OE2	35.6 ± 4.3 *	6.7 ± 0.3	63.1 ± 1.3	179.6 ± 14.5 *	72.6 ± 9.2 *	0.41 ± 0.02	21.8 ± 0.8

Notes: At least 40 plants were measured for each genotype. The plant and ear heights were measured from the ground level to the top of the tassel and to the node bearing the uppermost ear, respectively. The flowering time was calculated from the emergence to the shedding date. Values shown are the mean ± standard deviation. Asterisks (*) indicate statistically significant differences between the indicated genotypes and the WT by the Student’s *t*-test analysis (*p* < 0.01). PHT = plant height; EHT = ear height; WT = wild type.

## Data Availability

These sequence data have been submitted to the GeneBank database under accession numbers SUB11505871 and SUB11476127 (http://www.ncbi.nlm.nih.gov, accessed on 9 May 2022).

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
