# Peer review of "SDG102, a H3K36-Methyltransferase-Encoding Gene, Plays Pleiotropic Roles in Growth and Development of Maize (Zea mays L.)"

_ijms, 2022, doi:10.3390/ijms23137458_

Round 1

Reviewer 1 Report

In this study, the authors tried to demonstrate that SDG102, a close homolog of rice SDG708 and Arabidopsis SDG26, is a an H3K36 methyltransferase and regulates flowering time and other biological processes.

First, there is lack of clear novelty and breakthrough in this study.

Second, this manuscript is not well written that several figures and main text don’t always match and are not reader-friendly enough.

Third, the authors did a poor job analyzing and presenting the RNAseq data. There is not enough strong evidence supporting what the authors claim in the text.

Line 2: SDG102

Line 94: please give the full name of SDG protein when used the first time and then the term SDG can be used repeatedly.

Line 115-120: Figure S1 does not match what is described in Result section 2.1.

Line 122: transferases in rice (Oryza sativa) and Arabidopsis thaliana, respectively.

Line 124: Generation of SDG102 RNAi and overexpression lines

Line 126: The first approach was RNA interference that transgenic plants with downregulated SDG102 was generated.

Line 127: The second approach to generate SDG102 knockout transformants using CRISPR/Cas9.

Line 134: transgenic maize plants overexpressing SDG102 under constitutive 35S promoter were generated by pollen-mediated transformation

Line 143: overexpression line.

Line 152: Show gene structure of SDG102 and describe how may exons and introns SDG102 has before saying CRISPR/Cas9 constructs target third and seventh exons of SDG102 gene.

Line 164: Table S1 shows the primers used in this study instead of transformation rate.

Line 177: Result section 2.4, not 1.4.

Line 179: SDG102.

Line 180: Use AS1 or RNAi line.

Line 185: are referred to

Line 199: its width.

Line 200: SDG102.

Line 209: SGD102.

Line 229: “negative CK” instead of “Netative CK” in Figure 3m.

Line 253: GO term analysis was shown in Figure S3, not S2.

Line 266: Supplemental tables don’t match what the authors describe here in Result section 2.6 and the authors need to provide the expression level (ex: FPKM) of each gene (or at least DEGs) in WT, AS1, and OE lines. If the authors need to compare and discuss the expression levels of certain genes, make a figure to present that result which is more reader-friendly. The authors did a poor job analyzing and presenting the RNAseq data here. Major revision is required.

Author Response

ijms-1742757

Reviewer 1

In this study, the authors tried to demonstrate that SDG102, a close homolog of rice SDG708 and Arabidopsis SDG26, is a an H3K36 methyltransferase and regulates flowering time and other biological processes.

 First, there is lack of clear novelty and breakthrough in this study.

Response: Thank you for the common.  We have surveyed the relevant literature, and not able to find reports regarding the functions and roles of histone lysine methylation that caused by SDG proteins in maize growth and development. One possible reason might be due to the lack of mutants with obvious phenotypes.

Second, this manuscript is not well written that several figures and main text don’t always match and are not reader-friendly enough.

Response: Thank for the kind reminding and suggestions. We have rearranged Figure 1, Figure 3, Figure 5, Table S1 and corresponding main text according to the reviewer’s suggestion.

Third, the authors did a poor job analyzing and presenting the RNAseq data. There is not enough strong evidence supporting what the authors claim in the text.

Response: We thank for the commons.  We did present additional RT-qPCR results to support the results. Subsequently, we removed some unrelated genes expression and inaccurate expression data from this study. We are grateful for reviewer’s critics.

 Line 2: SDG102

Line 94: please give the full name of SDG protein when used the first time and then the term SDG can be used repeatedly.

Response: We thank for the commons. We modified the text according to the suggestion.

Line 115-120: Figure S1 does not match what is described in Result section 2.1.

Response: We thank for the critics.  The text of Result section 2.1 has been edited according to reviewer’s suggestion.  

Line 122: transferases in rice (Oryza sativa) and Arabidopsis thaliana, respectively.

Response: We thank for the commons. We modified the text according to the suggestion.

Line 124: Generation of SDG102 RNAi and overexpression lines

Response: We thank for the commons. We modified the text according to the suggestion and changed ‘upregulation’ to ‘overexpression’. SDG102 downregulation lines in this paper were obtained by both RNAi and CRISPR/Cas9. So SDG102 RNAi lines are not accurate expression here.

Line 126: The first approach was RNA interference that transgenic plants with downregulated SDG102 was generated.

Response: We thank for the commons. We modified the text according to the suggestion.

Line 127: The second approach to generate SDG102 knockout transformants using CRISPR/Cas9.

Response: We thank for the commons. We modified the text according to the suggestion.

Line 134: transgenic maize plants overexpressing SDG102 under constitutive 35S promoter were generated by pollen-mediated transformation.

Response: We thank for the commons. We modified the text according to the suggestion.

Line 143: overexpression line.

Response: We thank for the commons. We modified the text according to the suggestion.

Line 152: Show gene structure of SDG102 and describe how may exons and introns SDG102has before saying CRISPR/Cas9 constructs target third and seventh exons of SDG102 gene.

Response: We thank for the commons. We modified the text according to the suggestion and provided link to get to get gene structure of SDG102 at Gramene.

Line 164: Table S1 shows the primers used in this study instead of transformation rate.

Response: We thank for the commons. We rearranged the supplement tables according to the sequence mentioned in the paper according to reviewer’s suggestion.

Line 177: Result section 2.4, not 1.4.

Response: We thank for the commons. We modified the text according to the suggestion.

Line 179: SDG102.

Response: We thank for the commons. We modified the text according to the suggestion.

Line 180: Use AS1 or RNAi line.

Response: We thank for the commons. We modified the text according to the suggestion.

Line 185: are referred to

Response: We thank for the commons. We modified the text according to the suggestion.

Line 199: its width.

Response: We thank for the commons. We modified the text according to the suggestion.

Line 200: SDG102.

Response: We thank for the commons. We modified the text according to the suggestion.

Line 209: SGD102.

Response: We thank for the commons. We modified the text according to the suggestion.

Line 229: “negative CK” instead of “Netative CK” in Figure 3m.

Response: We thank for the commons. We modified the text according to the suggestion.

Line 253: GO term analysis was shown in Figure S3, not S2.

Response: We thank for the commons. We modified the text according to the suggestion.

Line 266: Supplemental tables don’t match what the authors describe here in Result section 2.6 and the authors need to provide the expression level (ex: FPKM) of each gene (or at least DEGs) in WT, AS1, and OE lines. If the authors need to compare and discuss the expression levels of certain genes, make a figure to present that result which is more reader-friendly. The authors did a poor job analyzing and presenting the RNAseq data here. Major revision is required.

Response: We thank for the commons. We modified the text according to the suggestion and made it clear that RT-qPCR results are additional evidence to support the results after deleting unrelated genes with this study and inaccurate expression.

Reviewer 2 Report

Positive aspects of the review:

The article is interesting and very useful for fundamental biochemical plant genetics, as it is related to SDG102 is an H3K36 methyltransferase and affirmed SDG102 promoted flowering by activating the key flowering genes ZCN8 / ZCN7. Deciphering the intimate mechanisms of gene expression is still terra incognita in many aspects of molecular biology. In my opinion, the development is at a high scientific level and has a number of innovations in research of this kind, as detailed studies on maize of this scale are almost non-existent.

Negative aspects of the review:

1. The text of the manuscript lacks a clearly defined purpose of the study, which corresponds closely to the title of the article.

2. Some of the figures are not presented clearly enough. For example, in Figure 1, I recommend enlarging the scale of the original DNA electrophoregram photograph.

3. Please formulate the conclusions clearly and precisely at the end of the Discussion section or differentiate a new section Conclusions!

4. Although the study is about fundamental biochemical genetics, the authors could make some useful recommendations for practice.

Author Response

ijms-1742757

Reviewer 2

Positive aspects of the review

The article is interesting and very useful for fundamental biochemical plant genetics, as it is related to SDG102 is an H3K36 methyltransferase and affirmed SDG102 promoted flowering by activating the key flowering genes ZCN8 / ZCN7. Deciphering the intimate mechanisms of gene expression is still terra incognita in many aspects of molecular biology. In my opinion, the development is at a high scientific level and has a number of innovations in research of this kind, as detailed studies on maize of this scale are almost non-existent.

Response: We are very much appreciated for the common and encouragement. It would be our honor to continue further research based on this work to advance our understanding of SDG102 function with maize growth and development.

Negative aspects of the review:

  1. The text of the manuscript lacks a clearly defined purpose of the study, which corresponds closely to the title of the article.

Response: We thank for the kind common.  We amended line 14-17 attempting to provide a clear defined description of purpose of the study.

  1. Some of the figures are not presented clearly enough. For example, in Figure 1, I recommend enlarging the scale of the original DNA electrophoregram photograph.

Response: We thank for the kind common and suggestions.  Effort have been made to revised Figure 1, Figure 3, Figure 5 and Table S1 which are corresponding to main text in attempt to be reader friendly.

  1. Please formulate the conclusions clearly and precisely at the end of the Discussion section or differentiate a new section Conclusions!

Response: We thank for the kind common and suggestions. Discussion section has been edited and the redundant content has been removed base on the expert’s suggestions.

  1. Although the study is about fundamental biochemical genetics, the authors could make some useful recommendations for practice.

Response: We thank for the kind common and suggestions. Suggestion in maize breeding practice with altered level of SDG 102 gene expression levels could give improved agronomic traits have been made.

Reviewer 3 Report

1.In the introduction, a lot of content are unnecessary,such as line46-49 and 71-82,which are no connection with the SDG102 and its target genes. More information and function of SDG family flowering time and other processes need to be introduced here just like you did in the discussion. Also in discussion, much content is redundant. This is where you should discuss your own research result and look forward to the future research direction in addition to summary.

2. If the focus of this paper is on the effect of SDG102 on flowering time in maize, more pictures of flower-related phenotypes are needed. Since deletion of SDG102 causes maize leaves to turn yellow in the early stages of growth, whether the photosynthesis and photosynthesis-related genes are affected?

3. The conclusion of SDG102 promotes flowering in maize by depositing H3K36me2/3 need to be further verification. How the H3K36me2 and H3K36me3 levels in most regions of the key flower genes ZCN8/ZCN7 chromatin compared to WT? whether consist with the flowering time? Since the H3K36me2 level of ZCN8/ZCN7 also has changed, why only the H3K36me3 level of SDG102 on global histone has been verified?

4. The gene mentioned at line 274-279, ZCN12,ZCN15, ZCN18 and ZCN26, RT-qPCR of these expression need to be done to support your point. 5.More details should be noted .such as Line 177,there should be 2.4 not 1.4;line 239,scale bar in k and l not j and k; line 291, b and c have bigger fonts. In reference 30, The title of the publication is Plant Physio. And in reference, some title of publications are italic.         

Author Response

ijms-1742757

Reviewer 3

  1. In the introduction, a lot of content are unnecessary,such as line46-49 and 71-82,which are no connection with the SDG102 and its target genes. More information and function of SDG family flowering time and other processes need to be introduced here just like you did in the discussion. Also in discussion, much content is redundant. This is where you should discuss your own research result and look forward to the future research direction in addition to summary.

Response: We are very grateful for the commons and suggestion. The unnecessary content has been removed, information & function of SDG family flowering time and other processes are emphasised in the introduction section. Any unrelated content in discussion was deleted according to the expert’s suggestions.

  1. If the focus of this paper is on the effect of SDG102 on flowering time in maize, more pictures of flower-related phenotypes are needed. Since deletion of SDG102 causes maize leaves to turn yellow in the early stages of growth, whether the photosynthesis and photosynthesis-related genes are affected?

Response: We are very grateful for the commons and suggestion. As recommended by expert, the picture of flower-related in figure 3m was added. Pigment content in leaves of SDG102 RNAi plants showed sharp deceasing as demonstrated in figure 3i. We’re evaluating whether the photosynthesis and photosynthesis-related genes were affected, that also included cob color changing cause of overexpression lines. Subsequent research and results will be reported in next paper.

  1. The conclusion of SDG102 promotes flowering in maize by depositing H3K36me2/3 need to be further verification. How the H3K36me2 and H3K36me3 levels in most regions of the key flower genes ZCN8/ZCN7 chromatin compared to WT? whether consist with the flowering time? Since the H3K36me2 level of ZCN8/ZCN7 also has changed, why only the H3K36me3 level of SDG102 on global histone has been verified?

Response: We are very grateful for the commons and suggestion. As shown in Figures 4b and c, there are significant reduction of H3K36me2 and H3K36me3 level in most regions of the key flower genes ZCN8/ZCN7 chromatin in SDG102 RNAi line, when we compared these with WT plants. The late flowering in SDG102 RNAi line was observed that are shown in Table 1 and Figure 3m.

We just got a preliminary conclusion of SDG102 promotes flowering in maize by depositing H3K36me2/3 and more detailed work are carrying out by another team and will report in next manuscript.

  1. The gene mentioned at line 274-279, ZCN12, ZCN15, ZCN18 and ZCN26, RT-qPCR of these expression need to be done to support your point.

Response: We are very grateful for the commons and suggestion. As expert’s suggested, these four genes are not closely related to the conclusions of this manuscript, we deleted this part from the section.

  1. More details should be noted .such as Line 177,there should be 2.4 not 1.4;line 239,scale bar in k and l not j and k; line 291, b and c have bigger fonts. In reference 30, The title of the publication is Plant Physio. And in reference, some title of publications are italic.     

Response: We are very grateful for the commons and suggestion We revised the manuscript based on reviewer’s suggestion and paid extra attention to the format.     

Round 2

Reviewer 3 Report

looking forward your next paper.

Author Response

Dear Reviewer, 

Thank you for the kind words and encouragement. It would be our honour to carry the subsequent studies in this field and continue the journal seeking understanding of functions of gene like SDG102 in maize growth and development. 

All the best,

Jian